# Protocol for a multicentre cross-sectional, longitudinal ambulatory clinical trial in rheumatoid arthritis and Parkinson's disease patients analysing the relation between the gut microbiome, fasting and immune status in Germany (ExpoBiome)

Bérénice Hansen,[1] Cédric C Laczny,[1] Velma T E Aho,[1] Audrey Frachet-Bour,[1] Janine Habier,[1] Marek Ostaszewski,[1] Andreas Michalsen,[2,3] Etienne Hanslian,[2,3] Daniela A Koppold,[2,3] Anika M Hartmann ![ORCID],[4,5] Nico Steckhan,[2,6] Brit Mollenhauer,[7,8] Sebastian Schade ![ORCID],[7,8] Kirsten Roomp,[1] Jochen G Schneider ![ORCID],[1,9] Paul Wilmes ![ORCID][1,10]

JGS and PW contributed equally.

For numbered affiliations see end of article.

**Correspondence to**
Jochen G Schneider;
jochen.schneider@uni.lu and
Professor Paul Wilmes;
paul.wilmes@uni.lu

## ABSTRACT

**Introduction** Chronic inflammatory diseases like rheumatoid arthritis (RA) and neurodegenerative disorders like Parkinson's disease (PD) have recently been associated with a decreased diversity in the gut microbiome, emerging as key driver of various diseases. The specific interactions between gut-borne microorganisms and host pathophysiology remain largely unclear. The microbiome can be modulated by interventions comprising nutrition.

The aim of our clinical study is to (1) examine effects of prolonged fasting (PF) and time-restricted eating (TRE) on the outcome parameters and the immunophenotypes of RA and PD with (2) special consideration of microbial taxa and molecules associated with changes expected in (1), and (3) identify factors impacting the disease course and treatment by in-depth screening of microorganisms and molecules in personalised HuMiX gut-on-chip models, to identify novel targets for anti-inflammatory therapy.

**Methods and analysis** This trial is an open-label, multicentre, controlled clinical trial consisting of a cross-sectional and a longitudinal study. A total of 180 patients is recruited. For the cross-sectional study, 60 patients with PD, 60 patients with RA and 60 healthy controls are recruited at two different, specialised clinical sites. For the longitudinal part, 30 patients with PD and 30 patients with RA undergo 5–7 days of PF followed by TRE (16:8) for a period of 12 months. One baseline visit takes place before the PF intervention and 10 follow-up visits will follow over a period of 12 months (April 2021 to November 2023).

**Ethics and dissemination** Ethical approval was obtained to plan and conduct the trial from the institutional review board of the Charité-Universitätsmedizin Berlin (EA1/204/19), the ethics committee of the state medical association (Landesärztekammer) of Hessen (2021–2230-zvBO) and the Ethics Review Panel (ERP) of the University of Luxembourg (ERP 21–001A ExpoBiome). The results of this study will be disseminated through peer-reviewed publications, scientific presentations and social media.

**Trial registration number** NCT04847011.

## STRENGTHS AND LIMITATIONS OF THIS STUDY

⇒ The participants of the longitudinal study will be closely monitored for 12 months and routine blood parameters as well as anthropometric data and questionnaires will be precisely documented.

⇒ This study will identify novel microbiome-derived common and disease-associated molecules involved in immune system modulation in two major chronic diseases: rheumatoid arthritis (RA) and Parkinson's disease (PD).

⇒ This study aims at also identifying novel targeted pathways to control chronic inflammatory conditions in the future.

⇒ A limitation is the heterogeneity of the cohorts regarding age and sex, which is due to the prevalence of the diseases: RA is more common in women, while PD is more common in men and has a later disease onset.

⇒ A bias exists in choosing RA and PD as chronic disorders to study immunophenotypes although generalisable results are targeted.

## INTRODUCTION

The human microbiome is emerging as a key driver of various diseases through its complex of distinct yet connected biomolecules (referred to as the '*expobiome*').[1 2] The expobiome comprises a diverse set of nucleic acids, polypeptides and metabolites which, in the gut alone, are present in substantial concentrations.[1] However, the specific

interactions between gut-borne microorganisms and host (patho)physiology remain largely unknown. Although host genetics shape the composition of the gut microbiome, the latter is particularly influenced by non-genetic factors such as lifestyle and diet.[3 4] Therefore, the microbiome is a plausible target to modify health outcomes.

Individuals suffering from chronic diseases, including autoimmune, metabolic and neurodegenerative diseases as well as cancer, often present alterations in their gut microbiome composition. These shifts are typically characterised by an overgrowth of one or several microbial species with likely adverse effects as well as a decrease in beneficial taxa.[5] Such imbalances are referred to as dysbiosis. Although structural microbiome changes are clearly detectable, the mechanistic or functional consequences of dysbiosis are still largely unknown. However, they may result in dysregulated interactions with the immune system.[6] Considering the intricacy of the immune system, the question arises whether the observed microbiome changes are cause or consequence of disease. This implies that, in addition to the genetic predisposition of the host, the gut microbiome needs to be considered a potential pathogenic factor or major driver of disease onset and course.[3 4]

Rheumatoid arthritis (RA) and Parkinson's disease (PD) are two specific examples representing dysregulated microbiome-immune system interactions.[7 8] RA is a multifactorial, chronic and systemic autoimmune disease, primarily affecting the lining of the synovial joints with a higher risk and younger age for disease onset in women and a global prevalence of 1%.[9 10] The exact disease pathogenesis is still unclear and no cure for RA currently exists. In addition to the common local articular symptoms of RA, systemic comorbidities can affect the vasculature, metabolism and bones.[11] Besides various environmental risk factors, for example, smoking and a Western diet, the host microbiome is associated with the pathophysiology of the disease.[12] The diversity of the gut microbiome has been reported to be decreased in individuals with RA, compared with the general population, and is correlated with disease duration, activity and autoantibody levels.[13 14] Studies in murine models also report that autoimmune arthritis is strongly attenuated under germ-free conditions.[15] The introduction of specific bacteria, for example, segmented filamentous bacteria, into germ-free animals or oral infection with *Porphyromonas gingivalis* drive autoimmune arthritis through activation of T helper cells.[15] Several different taxa, including *Prevotella copri*, *Lactobacillus* spp and *Colinsella* spp are enriched in the gut microbiome of patients with RA and correlate positively with disease markers, for example, immunoglobulins IgA and IgG, while other taxa like *Haemophilus* spp and *Faecalibacterium* spp are typically found at lower abundances in patients with RA compared with healthy individuals.[13 16 17] Alterations of the gut microbiome may, therefore have an important impact on RA pathophysiology.[12]

PD affects 0.4%–2% of the population over 65 years worldwide and is the second most common progressive neurodegenerative disease with men being 1.5 times more likely to be affected than women.[18] Cardinal symptoms are not only motor deficiencies such as tremor and rigidity but also include a wide range of non-motor symptoms, such as hyposmia, depression, insomnia or cognitive impairment, severely impacting patients' quality of life.[19] Aggregations of the protein a-synuclein in the dopaminergic substantia nigra represent the main neuropathological manifestations.[20] PD-associated loss of dopaminergic neurons involves mechanisms of inflammatory and autoimmune responses with microglial activity as a major driver.[21] Dysbiosis of the gut microbiome has been associated with the characteristic motor deficits and pathophysiological changes in the enteric and central nervous systems in animal studies. Increased relative abundances of the genera *Akkermansia*, *Bifidobacterium*, *Lactobacillus* and *Methanobrevibacter* and decreased abundances in *Faecalibacterium* and *Roseburia* have been reported.[22 23] Two recently published clinical trials with prebiotic supplementation in PD observed a shift in gut microbiome composition, an increase in short-chain fatty acids (SCFA) and a reduction in non-motor symptoms.[24 25] Most patients with PD suffer from gastrointestinal symptoms such as constipation and irritable bowel syndrome-like symptoms.[26] The gut-brain axis, for example, by-products produced by the gut microbiome, may contribute to the production of a-synuclein aggregates in the enteric nervous system.[27] In addition, increased intestinal permeability[28] as driver for enteric inflammation occurs in PD and substantiates a role of peripheral inflammation in the initiation and the progression of the disease.[29]

One factor with known major impact on the gut microbiome and on chronic diseases is diet.[7] Dietary approaches as fasting have already been used by Hippocrates in the fifth century before Christ and have been applied ever since by numerous medical schools to treat acute and chronic diseases.[30–32] Various practices of caloric restriction through fasting have repeatedly shown remarkable health benefits.[33 34] Maifeld *et al* found that a 5-day fast followed by a modified Dietary Approach to Stop Hypertension (DASH), with additional emphasis on plant-based and Mediterranean diets, reduced systolic blood pressure, body mass index (BMI) and the need for antihypertensive medications at 3 months post intervention compared with DASH alone.[35]

Furthermore, Choi *et al* demonstrated that cycles of a fasting-mimicking diet suppress autoimmunity and stimulate remyelination via oligodendrocyte regeneration in a murine experimental autoimmune encephalomyelitis model.[36] Jordan *et al* described a reduction in monocyte metabolic and inflammatory activity after a short-term fast and conclude that fasting attenuates chronic inflammatory diseases without compromising monocyte capacity for mobilisation during acute infectious inflammation and tissue repair.[37]

These improvements can, however, typically only be maintained for a limited period of time, and the symptoms can reappear after reintroduction of the patients'

standard diet. Hence, protocols to sustain these beneficial effects are of utmost importance. In mouse models of PD, intermittent fasting (IF) has led to several improvements including decreased excitotoxicity, reduced neurodegeneration and protection against autonomic dysfunction, motor and cognitive decline.[38]

IF and prolonged fasting (PF) may have potent immunomodulatory effects, which may partially be mediated by the gut microbiome and the fasting-induced alterations of the latter.[39] These microbial shifts include upregulation of not only *Akkermansia muciniphila*, *Bacteroides fragilis*, other *Bacteroides* spp, Proteobacteria and butyric acid producing *Lachnospiraceae* but also *Odoribacter*, which is negatively associated with blood pressure.[35 40] Interestingly, an overall decrease in the Firmicutes/Bacteroidetes ratio could be observed, a high ratio is commonly associated with several pathologies, including RA.[41]

A potential mechanism underlying the observed beneficial effects induced by dietary interventions might be a direct gut microbiome–immune system interaction by pattern recognition. The microbiome can regulate the intestinal innate immune system by modulating toll-like receptor expression on immunosensor cell surface through microbe-associated molecular patterns, which can consequently trigger cytokine production and upregulation of molecules on antigen presenting cells, leading to activation of T cells.[42] Therefore, a change in gut microbiome composition can lead to different outcomes in immune signalling pathways and either favour or suppress inflammation and autoimmunity.

The impact and importance of the gut microbiome on human physiology and its potential modifications by nutrition and dietary patterns have been underestimated for centuries.[43] Reasons may include missing standardised therapeutic protocols, the interindividual variability not only in the response to fasting, lack of knowledge about possible adverse effects and difficulties in the interpretation of underlying mechanisms seen in clinical trials but also in the comparably low potential for achieving economic revenue or scientific impact.[8]

Modern experimental approaches and computational integration allow a multilayer analysis of digestive processes in low caloric settings, including the gut microbiome.[44] These technological developments also permit a closer investigation of the link between the immune system and severe caloric restriction.

To our knowledge, no clinical trials have been investigating the connection between IF or PF and PD in humans so far.[38] Our study aims to elucidate the causal relationship between the gut microbiome and the immune system. To do so, we will use analyses of the molecular basis of human–microbiome interactions enabled by high throughput methodologies such as the combination of metagenomics (MG), metatranscriptomics (MT) and metaproteomics. Moreover, we are aiming at identifying new genes, proteins, metabolites and host pathways facilitating the development of novel diagnostic and therapeutic tools.[45 46]

## METHODS AND ANALYSIS
### Study objectives
The first objective of the study is to define specific gut microbiome-derived molecules in RA and PD, compared with healthy individuals, and relate this information to the immunophenotypes of the individuals. The second objective is to identify and track common and disease-specific molecular signatures to predict the outcome of a gut microbiome-targeted therapeutic intervention, here fasting, on inflammation-driven symptoms in RA and PD. The third objective of the study is to identify and validate microbiome-derived effector molecules, which downregulate pro-inflammatory innate and adaptive immune pathways.

### Study design
The ExpoBiome cohort consists of 180 adult individuals, meeting the exclusion and inclusion criteria (table 1), for the cross-sectional study (objectives 1 and 3) and 60 adult individuals for the longitudinal study (objectives 2 and 3). There are five different arms in total: (1) RA—cross-sectional arm (60 patients), (2) PD—cross-sectional arm (60 patients) and (3) healthy controls—cross-sectional arm (60 patients), (4) RA—longitudinal arm (30 patients), (5) PD—longitudinal arm (30 patients) (figure 1).

At the first visit (T0), patients answer several questionnaires, and blood, urine, saliva and stool samples are obtained (box 1).

The longitudinal arms (4) and (5) undergo a 5–7 day PF with a dietary energy supply of maximum 350–400 kcal per day with vegetable or grain broths as well as fresh vegetable juices.[31 40] After the PF, the longitudinal arms follow a dietary regimen, including the concept of time-restricted eating (TRE) for a period of 12 months following the 16:8 pattern.[47] This means that food intake is allowed ad libitum for 8 hours, followed by 16 hours of fasting, where no food should be consumed. The intake of non-caloric beverages, for example, water, unsweetened tea or coffee is, however, allowed. The participants attend one follow-up visit (T2) during the PF and nine follow-up visits during the 12 months of TRE (figure 1).

### Patient and public involvement
Feedback of patients during former clinical trials at the study centre in Berlin was integrated in the planning and design of the fasting intervention of this study. Patients are not involved in the conduct, reporting or dissemination plans of this research.

### Recruitment and randomisation
Patients are recruited by the specialised sites via different sources, for example, by direct referral from either a physician at the Immanuel Hospital Berlin and the outpatient department of the Institute of Social Medicine, Epidemiology and Health Economics at Charité-Universitätsmedizin Berlin, or the Paracelsus-Elena Clinic

**Table 1** Inclusion and exclusion criteria

| Inclusion criteria | Exclusion criteria |
| --- | --- |
| ► Age 18–79<br>► One of the following diagnoses: rheumatoid arthritis (first diagnosis>6 weeks ago), Parkinson's disease OR healthy volunteer<br>► Control ('healthy') individuals must be without any evidence of active known or treated RA, without any evidence of active, known or treated central nervous system disease, and without a known family history of idiopathic PD<br>► Control individuals should match the RA or PD individuals as closely as possible (sex, age, education)<br>► Present written declaration of consent<br>► Ability to understand the patient information and willingness to sign the consent form<br>► Consent to specimen collection and specimen use | ► Gout or proven bacterial arthritis<br>► Participation in another study<br>► Existing/current eating disorder (bulimia nervosa, anorexia nervosa) within the past 5 years<br>► Severe internal disease (eg, kidney deficiency with creatinine >2 mg/dL)<br>► Existing vegan diet or fasting during the last 6 months<br>► Presence or suspicion of atypical PD (eg, early dementia, early autonomous dysfunction)<br>► Diagnosis of chronic inflammatory bowel diseases, coeliac disease or colorectal cancer according to the guidelines of the German Society of Gastroenterology<br>► Use of anti-psychotic drugs<br>► Antibiotic use during the previous 12 months<br>► Start of novel therapy with disease-modifying anti-rheumatic drugs<br>► Pregnancy or breastfeeding women<br>► Contraindication for additional blood draws (eg, haemoglobin <10)<br>► BMI<18.5<br>► Psychiatric illness that limits understanding of the examination protocol (unable to consent) |

BMI, body mass index; PD, Parkinson's disease; RA, rheumatoid arthritis.

in Kassel, or by non-personal advertising strategies (eg, flyers or social media).

For PD, the patients are screened by an experienced movement disorders specialist for featuring at least two of resting tremor, bradykinesia and rigidity according to the United Kingdom Parkinson's Disease Society Brain Bank criteria.[48] Additionally, patients must show evidence of a dopaminergic deficit, either with DaTScan imaging or with a clear response to dopaminergic drugs. Motor and non-motor symptoms are assessed with the MDS-UPDRS (part I–IV), including the Hoehn and Yahr (severity) scale.[49] Additional PD-specific scales as Parkinson's Disease Sleep Scale-2, Parkinson's Disease Questionnaire-39, Non-Motor Symptoms Questionnaire and Non-Motor Symptoms Scale are used.

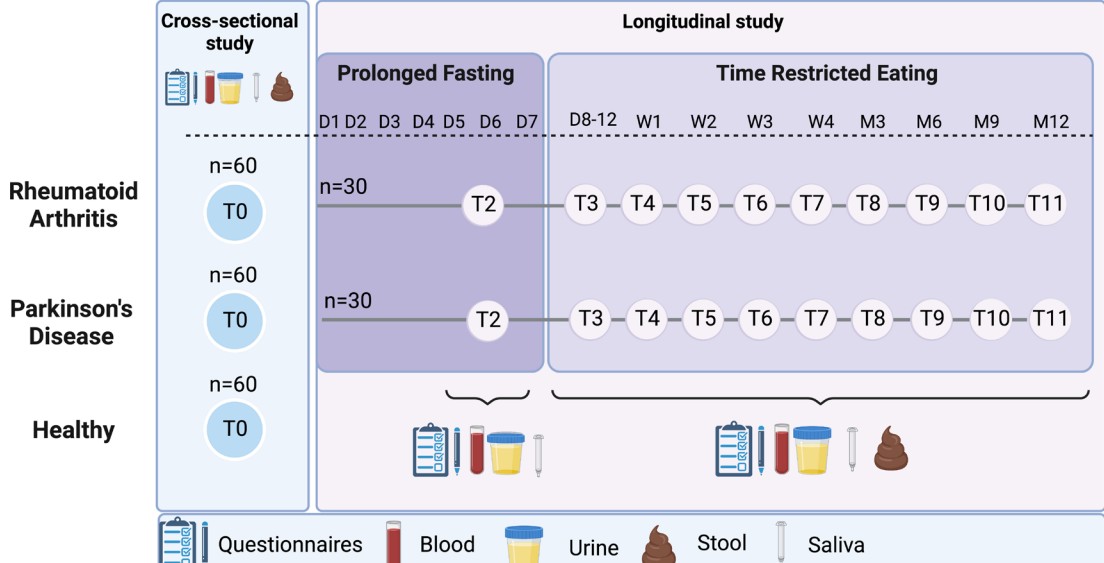

**Figure 1** Study design. This figure illustrates the study design with five different arms in total, two of which continue with the longitudinal part of the study. Visits take place at the clinical sites at each timepoint and include the collection of the displayed samples. This image was generated using Biorender software (http://www.biorender.com). D, day; M, month; T, timepoint; W, week.

## Box 1  Sampling procedures

**Biochemical samples and procedures**
⇒ Blood (123 mL at T0, 23 mL at T2–T11).
⇒ Stool collection (2 mL at T0 and T3–T11).
⇒ Saliva collection (3.5 mL at T0–T11).
⇒ Midstream urine (50 mL at T0–T11).

**Questionnaires**
Disease-specific
Parkinson's disease (PD)
⇒ Disease Activity Score.[72]
⇒ Parkinson's Disease Sleep Scale-2.[73]
⇒ Parkinson's Disease Questionnaire-39.[74]
⇒ Simplified Disease Index Score.[75]
⇒ Funktionsfragebogen Hannover.[76]
⇒ Movement Disorder Society Unified PD Rating Scale.[77]
⇒ Non-Motor Symptoms Questionnaire.[78]
⇒ Non-Motor Symptoms Scale.[79]
Rheumatoid arthritis
⇒ Disease Activity Score.[75]
⇒ Non-Motor Symptoms Questionnaire.[78]
⇒ Funktionsfragebogen Hannover.[76]
Dietary behaviour and lifestyle
⇒ Fasting experience, expectation, and intervention.
⇒ Lifestyle.
⇒ 24H-Food-recall.
⇒ Food Frequency Questionnaire.
General health and well-being
⇒ Health Assessment Questionnaire.[80]
⇒ Bristol Stool Scale.[81]
⇒ Quality of Life questionnaire.[82]
⇒ Hospital Anxiety and Depression Scale.[83]
⇒ Profile of Mood States.[84]

For patients with RA, the diagnosis has been made prior to the study by an experienced rheumatologist according to the European League Against Rheumatism criteria.[50] All clinical stages of RA will be included. We excluded patients with a BMI <18.5, as this indicates underweight, and fasting is not recommended. We did, however, not include an upper limit as fasting might be especially beneficial for patients with a BMI >24.9 and more than 60% of patients with RA are classified as overweight or obese.[51] For comorbidities, we excluded mainly diseases which are known to interfere with the gut microbiome and might be potential confounders.

The chosen exclusion criteria will optimise the pairing process of healthy controls and patients with either RA or PD. However, as we have two diseases with different anthropometric characteristics (including age, gender, BMI) and only one control group, adding additional inclusion and exclusion criteria in the recruitment process would compromise on optimised matching. Furthermore, for the longitudinal part of the study, each patient will serve as his/her own control over time. Participants meeting all the inclusion and no exclusion criteria (table 1) are assigned to their respective groups (RA, PD or healthy control) (figure 1) for the cross-sectional study after written informed consent.

Half of the patients from the RA group and half of the patients from the PD group are selected to take part in the longitudinal part of the study, including the fasting intervention according to their availability for all 11 visits and their willingness to follow TRE over 12 months. This study is an open-label trial, as blinding is not feasible in fasting interventions.

### Fasting dietary counselling
The fasting group is closely monitored by nutritionists trained in fasting therapy, backed up by physicians experienced in fasting, from the Charité—Universitätsmedizin Berlin and the Paracelsus-Elena Clinic to ensure a uniform implementation of the fasting guidelines and the well-being of the study participants. The monitoring consists of several in person and virtual meetings, which held individually or in group settings. Five meetings including the visits T0 and T2 during the fasting week as well as a group meeting after PF to ensure a well-managed start to the TRE phase takes place. Group sessions are standardised using a preset deck of slides to be discussed during the group meetings with only minor disease-related differences between the PD and RA groups. All longitudinal participants receive a study-specific script with information on fasting procedures. Although the adherence of the patients cannot be profoundly controlled in the ambulatory setting, the blood samples will allow us to have additional insight into the nutritional habits as well as the fasting state of the patients on the day of the visit (blood glucose levels).

### Medication
The medical treatments of the patients are monitored and documented with every clinical visit. The fasting intervention might necessitate temporary adjustments of several medications, for example, antidiabetic and antihypertensive drugs as insulin levels and hypertension will be reduced due to lack of food intake.[31]

### Data collection
Sample and data collection are performed at the two clinical sites, Charité—Universitätsmedizin Berlin and Paracelsus-Elena Clinic (box 1).

#### Anthropometric data and questionnaires
The electronic data capture system REDCap,[52] a secure web-based application, is used to record all individual specific data. All data are stored on a secure server infrastructure at the host institution in Luxembourg. Weight, height, BMI, heart rate and blood pressure in sitting and standing position as well as waist–hip–ratio are determined at every visit. Dietary behaviour, sociodemographic measurements (age, sex, education level, employment status, marital status), family history, current and previous illness and comorbidities and current medications as well as disease-specific data, questionnaires about the well-being of the patients and data on the behavioural factors are collected at baseline, T6 (week 3), T9 (month 6) and T11 (month 12) (box 1). Questionnaires (24 hour-Food

**Table 2** Routine blood parameters measured at each timepoint (T0 for cross-sectional study, T0–T11 for longitudinal study)

| Haematology—EDTA-blood | Clinical chemistry—serum |
|---|---|
| Basophils, % | Albumin |
| Basophils, abs. | Alanine Transaminase (ALT), 37°C |
| Eosinophils, % | Alkaline phosphatase, 37°C |
| Eosinophils, abs. | Aspartate Transferase (AST), 37°C |
| Erythrocytes | Bilirubin, total |
| Haematocrit | Cholinesterase |
| Haemoglobin | Cholesterol |
| HbA1c | Creatinine |
| Leucocytes | high-sensitivity C-reactive protein (hs-CRP) |
| Lymphocytes, % | Glucose, serum |
| Lymphocytes, abs. | Gamma-GT, 37°C |
| Mean corpuscular hemoglobin (MCH) | High density lipoprotein (HDL)-cholesterol |
| Mean corpuscular hemoglobin concentration (MCHC) | Low density lipoprotein (LDL)-cholesterol |
| Mean corpuscular volume (MCV) | Potassium |
| Monocytes, % | Sodium |
| Monocytes, abs. | Total protein |
| Neutrophils, % | Triglycerides |
| Neutrophils, abs. | Uric acid |
| Platelets | Urea/Blood Urea Nitrogen |
| Red cell distribution width (RDW) | Proteins—serum |
| Reticulocytes | Rheumatoid factor H 35.9 |
| Reticulocytes | Hormones—serum |
| Reticulocytes, abs. | Insulin |
| | Thyroid stimulating hormone (basal) |

Recall, Bristol Stool Scale) are answered at all visits by the study participants. Data storage, analysis and exchange are done only in pseudonymised fashion. The nutritional data are analysed using the Nutrilog V.3.20 software (Nutrilog SAS, Marans).

## Blood samples and parameters

Blood samples are collected at each visit and immediately used for peripheral blood mononuclear cell (PBMC) isolation (T0), analysis by the study laboratory and centrifugation to freeze plasma samples at −80°C (T0–T11). A clinical standard laboratory report is generated after every visit for each study participant (table 2). In addition to routine blood parameters, anticitrullinated protein antibody, zonulin, fatty acid binding protein 2 and calprotectin levels are measured. Aliquots are securely stored to account for novel observations and testing of hypotheses.

## Stool, urine and saliva samples

The samples listed in box 1 are collected at each visit, except for stool samples on T2 (fasting week) and immediately frozen and stored at −80°C. Stool characteristics are recorded at the time of the sampling. Faecal samples represent the main sample type for resolving the dynamic processes driven by microbiome in the gut. Also, as the gut microbiome is prone to diurnal fluctuations, the stool samples are collected in the morning, as far as possible.

## Methods applied to samples

### Biomolecular extractions

The collected stool samples undergo a biomolecular extraction procedure to allow isolation of concomitant DNA, RNA, proteins, peptides and metabolites from single, unique faecal water samples; this process involves cryomilling the samples in liquid nitrogen, disassociating metabolites from membrane and cell wall components in a solvent mixture of methanol, chloroform and water and lastly proteins and RNA extraction by a methanol/chloroform and phenol buffer.[53 54] Faecal water is recovered following centrifugation and filtration, at low speed or low flow, respectively, to avoid cell lysis. Nucleic acids are preserved by the addition of ribonuclease inhibitors and isolated by silica column-based techniques. This protocol involves the use of a robotic platform, ensuring a higher level of standardisation and reproducibility.[2]

### Coupled MG and MT analyses

Prior to sequencing library preparation, internal standards are introduced to obtain quantitative sequencing data.[55] Contamination-free MG and MT data are generated, processed and analysed using the integrated meta-omics pipeline (IMP),[45] which incorporates preprocessing, assembly, gene annotation, mapping of reads, single nucleotide polymorphism calling, data normalisation as well as analyses of community structure and function in a fully reproducible software framework based on Docker. The MG and MT data are specifically screened for enrichments in genes and pathways with known immunogenic properties.[56] The extracellular biomolecules are linked to specific microbial populations based on the intracellular MG data.[57] In addition, the sequencing data are mapped against genomes of food components.[44] The quantitative data are also related to microbial population sizes to determine the contribution of the resolved microbial populations in stool to the extracellular DNA and RNA complements.[58]

### Metaproteomics

For the metaproteomic analyses, filtration is used to separate extracellular peptides from the obtained (poly) peptides. The resulting smaller fractions are then desalted and analysed without proteolytic digestion via liquid chromatography (LC) and mass spectrometry (MS) on

an EasyNano-LC coupled online to a QExactive-Plus mass spectrometer (ThermoScientific, Waltham). The identification of ribosomal peptides is done with an integrated catalogue of MG and MT data, while the non-ribosomal peptides are identified using different tools, that is, MyriMatch, DirecTag as well as CycloBranch.[45 59 60] The metaproteomic data also allow identification of extracellular (poly)peptides with possible pathogenic functions including protein misfolding and molecular mimicry.[61 62]

## Metabolomics

Metabolomic data are analysed using a combination of targeted and untargeted approaches.[44 54 63] This highlights the major metabolite classes produced by the gut microbiome with an effect on human physiology, including organic acids, SCFA, lipids, branched-chain fatty acids, branched-chain amino acids, vitamins, bile acids and neurotransmitters. Besides external compound calibration series for quantification and quality control samples to ensure data normalisation and data acquisition quality assessment, the metabolite extraction fluid is fortified with multiple internal standards to improve method precision and accuracy.[64 65] The data are compared with in-house databases and public mass spectral libraries to identify known metabolites. The metabolomic data complement the MG and MT data and, thus, allow further establishments of conclusive links to metabolic properties in the gut.

## Deep immune profiling

Deep immune profiling is done using a recently established and optimised panel of metal-labelled antibodies together with cytometry coupled to MS, the Maxpar Direct Immune Profiling System (MDIPA). This approach allows the simultaneous quantification of 38 parameters on single cells. Whole blood is stained with the MDIPA kit and stabilised with Proteomic stabiliser Prot-1 (501351694, Smart Tube, Las Vegas) before storage at −80°C. The quantified immune cells included in the MDIPA panel are CD3+, CD4+, CD8+, monocytes, dendritic cells, granulocytes, mucosal-associated invariant T cells (MAIT), T cells, natural killer (NK) and B cells.[66] Cytokine expression profiles are analysed on blood plasma using the Human Luminex performance Cytokine Panel (R&D Systems Europe, Abingdon), measuring CCL3, CCL4, CCL5, GM-CSF, IL-1b, IL-2, IL-4, IL-5, IL-6, IL-8, IL-10, IL-12p70, IL-13, IL-15, IL-18, IL-21, IL-27, IL-33, IFN-b, Galectin-1, IFN-g and TNF-a.[56]

## Gut-on-a-chip models

PBMCs isolated from T0 blood samples are co-cultured with gut-derived microbes under physiologically representative conditions using the gut-on-a-chip model HuMiX.[67] This model of the human gastrointestinal interface allows the investigation of the interactions between immune, epithelial and bacterial cells and specifically the response to fasting in personalised in vitro models.

## The expobiome MAP

The Expobiome Map (https://expobiome.lcsb.uni.lu) illustrates the diverse complex of microbial immunogenic molecules, including nucleic acids, (poly)peptides, structural molecules, and metabolites. The interactions between this "expobiome" and human immune pathways are encoded in the context of chronic diseases.[1] The ExpoBiome Map is visualised using the MINERVA Platform.[68] Clicking on different elements on the map reveals factors they affect and are affected by, allowing an easier navigation through the complex relationships between individual microbiome components in relation to human disease. The multi-omics data generated in the present study will be integrated with the Map.

## Exploratory analysis of novel host-microbiome interactions

Unknown non-ribosomal peptides or metabolite features are associated through correlation with transcripts, proteins, and metabolites. Extracellular DNA fragments, as well as transcripts, proteins and ribosomal peptides are linked to their genomic context by using IMP.[45] The data generated by the project will be connected and collated to existing, publicly available datasets.

## Outcome parameters

### Primary outcome

The primary endpoint of the study is the characterisation of the gut microbiome. The evaluation includes both between-group and within-group differences in the longitudinal study arms with the fasting intervention.

### Secondary outcome measures

Secondary outcomes include the identification of common and disease-specific molecular signatures and the characterisation of microbiome-derived effector molecules impacting the innate and adaptive immune pathways. Furthermore, several additional parameters mentioned in *Anthropometric data and questionnaires* are assessed over a period of 12 months.

## Sample size and power calculation

A power calculation using pilot MT data based on faecal extracellular RNA samples was performed to determine the number of subjects to be recruited for the ExpoBiome project. The obtained relative abundances of genera were used for the calculation of the required sample size per group. The power calculation was based on the algorithm as described by Tusher, Tibshirani, and Chu.[69] To achieve a power of 90% (at α=0.05), a total of 50 individuals per group (RA, PD, healthy controls) must be analysed. Considering any possible dropouts, 20% additional subjects are recruited, resulting in a total number of 180 individuals, that is, 60 per group. For the longitudinal part, a subset of 60 adult individuals (30 patients with PD and 30 patients with RA) are selected, based on their ability and willingness to participate in the longitudinal part of the study (12 months follow-up). The selected number of participants for the longitudinal study is based on feasibility due to the complexity and high costs of the

clinical trial. The total number of subjects in the longitudinal study can be smaller, as each individual serves as their own control.

## Adverse events

There are no major risks expected for participants. Minor common adverse effects of PF might include headaches, nausea, insomnia, back pain, dyspepsia and fatigue.[70] Any occurring adverse events are recorded at each visit in REDCap.[52] Serious adverse events are communicated to the study coordinator and principal investigator within 24 hours of their report.

## Data management, monitoring, analysis and evaluation of data

The study participants receive a study ID (pseudonym), which is used for all collected data. Self-administered questionnaires are directly recorded in REDCap. Participant files are kept for at least 10 years at the respective clinical sites.

Weekly meetings between the study team, the different clinical partners and the principal investigator ensure a close monitoring of the data. Any occurring adverse events or other issues are, thus handled immediately.

Different statistical tests are performed according to the nature of the data. A premature termination of the study is not envisaged; therefore, no interim analysis is done. Different correlation measures are applied, including Spearman correlation, mutual information on discretised data, distance correlation, maximum information criterion, local similarity analysis and the bioenv approach. Comparison across all omic levels allows identification of common and disease-specific signatures. Multivariate machine learning is used to link different data features to observed patterns. For additional confounding factors, especially in the cross-sectional study, multivariate statistical analysis will be performed. These factors will be accounted by including confounders in the analysis, for example, as covariate in the statistical models.

The longitudinal part of the study continues for a period of 12 months. After finalisation of this period, there is no follow-up of the participants. Interesting findings will be further validated using the existing sample set and analyses may be performed on additionally collected samples.

The Standard Protocol Items: Recommendations for Interventional Trials checklist was used to write this protocol.[71]

## Trial status

The recruitment for the ExpoBiome study started in April 2021 and is currently ongoing. All study participants should be recruited by the end of 2022. The sample collection will take place from April 2021 to November 2023.

## ETHICS AND DISSEMINATION

Ethical approval was obtained to plan and conduct the trial from the institutional review board of the Charité-Universitätsmedizin Berlin (EA1/204/19), the ethics committee of the state medical association (Landesärztekammer) of Hessen (2021–2230-zvBO) and the Ethics Review Panel (ERP) of the University of Luxembourg (ERP 21–001 A ExpoBiome). The results of this study will be disseminated through peer-reviewed publications, scientific presentations as well as press releases and social media postings (Twitter, LinkedIn). Study participants will be contacted and informed by the respective clinical sites about the outcome and results of the study, once the data analysis has been completed (dissemination phase).

**Author affiliations**
[1]Luxembourg Centre for Systems Biomedicine, University of Luxembourg, Esch-sur-Alzette, Luxembourg
[2]Institute for Social Medicine, Epidemiology and Health Economics, Charité Universitätsmedizin Berlin, Berlin, Germany
[3]Department of Internal and Integrative Medicine, Immanuel Hospital Berlin-Wannsee Branch, Berlin, Germany
[4]Institute of Social Medicine, Epidemiology and Health Economics, Charité Universitätsmedizin Berlin, Berlin, Germany
[5]Department of Dermatology, Venereology and Allergology, Charité Universitätsmedizin Berlin, Berlin, Germany
[6]Digital Health-Connected Healthcare, Hasso Plattner Institute, University of Potsdam, Potsdam, Germany
[7]Neurosurgery, University Medical Center Göttingen, Gottingen, Germany
[8]Movement disorders and Parkinson's Disease, Paracelsus-Kliniken Deutschland GmbH, Osnabruck, Germany
[9]Department of Internal Medicine and Psychiatry, Saarland University Hospital and Saarland University Faculty of Medicine, Homburg, Germany
[10]Department of Life Sciences and Medicine, University of Luxembourg, Esch-sur-Alzette, Luxembourg

**Acknowledgements** We thank Dr. Catharina Delebinski, Melanie Dell'Oro, Grit Langhans, Ursula Reuß, Maik Schröder and Nadine Sylvester for their support during the study.

**Contributors** Study design and protocol were done by BH, CCL, JGS, PW; the interventional concept was drawn by EH, DAK-L, AM, AMH, BM, SS, NS, JGS, PW; the clinical trial was designed and was conducted by EH, DAK-L, AM, AMH, BM, SS; the procured funding was provided by PW; the planning of high-throughput applications, statistical planning, sample size calculation and randomisation were defined by CCL, JGS, PW, KR; the initial draft of the manuscript and coordination of the editing process were performed by BH; the protocol preparation has been done by BH, AF-B, JH; the planning of the data analysis was done by CCL, JGS, PW, KR, VTEA, MO; all authors contributed equally with edits, comments and feedback, read and approved the final manuscript.

**Funding** This project has received funding from the European Research Council (ERC) under the European Union's Horizon 2020 research and innovation program (grant agreement number 863664). This work was supported by the Luxembourg National Research Fund (FNR) under grant PRIDE/11823097.

**Competing interests** None declared.

**Patient and public involvement** Patients and/or the public were not involved in the design, or conduct, or reporting, or dissemination plans of this research.

**Patient consent for publication** Not applicable.

**Provenance and peer review** Not commissioned; externally peer reviewed.

Not applicable.

**ORCID iDs**
Anika M Hartmann http://orcid.org/0000-0002-0135-9643

Sebastian Schade http://orcid.org/0000-0002-6316-6804
Jochen G Schneider http://orcid.org/0000-0003-2139-0602
Paul Wilmes http://orcid.org/0000-0002-6478-2924

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
