## [Reviewer comments · BMJ Open]

ARTICLE DETAILS

TITLE (PROVISIONAL)	Protocol for a multicentre cross-sectional, longitudinal ambulatory clinical trial in rheumatoid arthritis and Parkinson's disease patients analysing the relation between the gut microbiome, fasting and immune status in Germany (ExpoBiome)
AUTHORS	Hansen, Bérénice; Laczny, Cédric C.; Aho, Velma T.E.; Frachet-Bour, Audrey; Habier, Janine; Ostaszewski, Marek; Michalsen, Andreas; Hanslian, Etienne; Koppold-Liebscher, Daniela; Hartmann, Anika; Steckhan, Nico; Mollenhauer, Brit; Schade, Sebastian; Roomp, Kirsten; Schneider, Jochen; Wilmes, Paul

VERSION 1 – REVIEW

REVIEWER	El Menofy, Nagwan Al-Azhar University, Microbiology and immunology
REVIEW RETURNED	16-Feb-2023

GENERAL COMMENTS	Reviewing Report Protocol for a multicentre cross-sectional, longitudinal study in rheumatoid arthritis and Parkinson's disease patients analysing the relation between the gut microbiome, fasting and immune status (ExpoBiome) General comments The protocol conducted is very great study scientifically sound and elegantly presented. The study aims to elucidate the causal relationship between the gut microbiome and the immune system in rheumatoid arthritis and Parkinson's diseases enabled by high throughput methodologies in addition to identifying new genes, proteins, metabolites and host pathways facilitating the development of novel diagnostic and therapeutic tools. Structured comments • 1-Title Adequate• 2- Abstract• The time of the study should be clarified also in the abstract.• Please clarify No. of cases for cross-sectional and longitudinal studies• 3- Key words:• Please add additional relevant key words as Metagenomic, Metatranscriptomic, Metaproteomics, Metabolomics.• 4-Introduction• Introduction is adequate. Please, include some brand-new publications on the expiome topic.
--

	 • Please add two subtitles under strengths and limitations of the study 5- Methods and Analysis  • Some referencing errors and typos in methods part should be corrected as shown in the sticky notes in pdf file . • Please illuminate urine and saliva used for which analysis test. • In line 308 Biomolecular extractions, please clarify the procedure for isolation of DNA, RNA, ptn, etc from faecal water. • In the Exclusion criteria, did the authors exclude patients with sporadic colitis (due to food or other interferences)? • Please add the city for each device or kit (ThermoScientific, City, USA). As in line 337 • Generally in methods part, authors mentioned some detailed analysis methods as in Metaproteomics and Metabolomics protocols while undetailed information in other as in coupled metagenomic and metatranscriptomic analyses etc please . • I want to ask about the effect of drugs for treatment of both diseases (rheumatoid arthritis and Parkinson's) it does not mention by authors to study the effect of different drugs on microbiome composition in both diseases in cross-sectional and longitudinal study please clarify this important point. • 6-Outcomes  • This section is adequate. • 7- Discussion  • Authors mainly talk about the microbiome dysbiosis and the effect of diet and fasting on microbiome and ignore discussing the interaction between microbiome and immune system in rheumatoid arthritis and Parkinson's please discuss this point deeply with introduction of more relevant new references. Comments to the editor I endorse the acceptance of the manuscript after performing the modifications.
--	---

REVIEWER	Cosentino, Marco University of Insubria, Center for Research in Medical Pharmacology
REVIEW RETURNED	24-Apr-2023

GENERAL COMMENTS	The protocol describes a potentially interesting study on the effects of fasting on the gut microbiome and on immune parameters in RA and PD patients and in control subjects. After a cross-sectional analysis of the three groups, subgroups of RA and PD patients will be followed for 12 months during which they will practice prolonged fasting and time-restricted eating. The study is timely and relevant for the better understanding of the role of the relationship between immunity and the microbiome in the progression of autoimmune and inflammatory chronic diseases such as PD and RA. Some issues however should be addressed to improve the description. Of the study. Throughout the text, the different groups of subjects should be homogeneously labelled to allow clear understanding of the
---

	protocol procedure. At present they are numbered but with different numbers in the various sections. Inclusion criteria include the diagnosis of PD or RA, but the diagnostic criteria should be better clarified to ensure a correct recruitment. Exclusion criteria encompass antibiotic therapy as well as pregnancy and breast feeding. But what about such condition occurring during the study? Will those be reasons for exclusion? Why? Or why not? And what about high BMIs? While unlikely, they will be included? Why? And up to which values? Pairing of patients with controls is said to be based on sex, age and education. What about BMI? What about comorbidities? Sample size is calculated for the cross-sectional but not for the longitudinal part of the study. While this might be acceptable, it should be justified. What about any procedures to check subject adherence to the study protocol during the longitudinal phase? The protocol states that no additional follow up will be performed on any subjects after the end of the study. However no mention is made of any feed back information for participants about the study results. It would be polite and possibly useful to provide each person at least with a general idea of the outcomes of the study and whether it was successful and which will be the implications of the obtained results, etc.
--	---

VERSION 1 – AUTHOR RESPONSE

II. Reviewer 1 : Dr. Nagwan El Menofy, Al-Azhar University

Protocol for a multicentre cross-sectional, longitudinal study in rheumatoid arthritis and Parkinson's disease patients analysing the relation between the gut microbiome, fasting and immune status (ExpoBiome)

General comments

The protocol conducted is very great study scientifically sound and elegantly presented. The study aims to elucidate the causal relationship between the gut microbiome and the immune system in rheumatoid arthritis and Parkinson's diseases enabled by high throughput methodologies in addition to identifying new genes, proteins, metabolites and host pathways facilitating the development of novel diagnostic and therapeutic tools.

Structured comments.

Thank you very much for this positive comment on the design of the study and for the additional comments on the manuscript.

1. Title : Adequate

2. Abstract : The time of the study should be clarified also in the abstract.

Thank you for this comment, the sample collection takes place from April 2021 to November 2023. This information has been added to the abstract (line 73).

Please clarify No. of cases for cross-sectional and longitudinal studies

The number of participants for the different cohorts are as follows: 60 patients with PD, 60 patients with RA and 60 healthy controls for cross-sectional part; 30 patients with PD and 30 patients with RA for the longitudinal study (line 68).

Key words: Please add additional relevant key words as Metagenomic, Metatranscriptomic, Metaproteomics, Metabolomics.

We appreciate this valuable input, the key words have been added to the list in the manuscript. The adjusted keywords are as follows: Microbiome, fasting therapy, intermittent fasting, time restricted eating, chronic disease, rheumatoid arthritis, Parkinson's disease, nutrition, chronic diseases, ExpoBiome, inflammation, gut on a chip, HuMiX, immunophenotype, metagenomics, metatranscriptomics, metaproteomics, metabolomics (line 87).

3. Introduction : Introduction is adequate. Please, include some brand-new publications on the expiome topic.

We thank the reviewer for making this point and introduced the following new publications into our introduction (line 107):

- Kitamura, K., et al., Oral and Intestinal Bacterial Substances Associated with Disease Activities in Patients with Rheumatoid Arthritis: A Cross-Sectional Clinical Study. *J Immunol Res*, 2022. 2022: p. 6839356.
- Becker, A., et al., Effects of Resistant Starch on Symptoms, Fecal Markers, and Gut Microbiota in Parkinson's Disease - The RESISTA-PD Trial. *Genomics Proteomics Bioinformatics*, 2022. 20(2): p. 274-287.
- Hall, D.A., et al., An open label, non-randomized study assessing a prebiotic fiber intervention in a small cohort of Parkinson's disease participants. *Nat Commun*, 2023. 14(1): p. 926.

4. Please add two subtitles under strengths and limitations of the study

The strength and limitations part of the manuscript has been structured according to the BMJ Open guidelines, this part will be added as a table in the final publication and therefore subtitles are unfortunately not possible.

5. Methods and Analysis: Some referencing errors and typos in methods part should be corrected as shown in the sticky notes in pdf file.

Thank you for this comment, the reference errors originated from a figure that has been deleted from the main file and uploaded separately as requested by BMJ Open. The typo has been adjusted as well.

6. Please illuminate urine and saliva used for which analysis test.

The urine and saliva samples are collected for potential workup later. The exact analysis is not determined yet.

7. In line 308 Biomolecular extractions, please clarify the procedure for isolation of DNA, RNA, ptn, etc from faecal water.

We appreciate the interest in this procedure and adapted the paragraph accordingly. The isolation procedure has been explained in greater detail as follows: "This process involves cryo-milling the samples in liquid nitrogen, disassociating metabolites from membrane and cell wall components in a solvent mixture of methanol, chloroform and water and lastly proteins and RNA extraction by a methanol/chloroform and phenol buffer" (line 509).

8. In the Exclusion criteria, did the authors exclude patients with sporadic colitis (due to food or other interferences)?

Only patients with chronic inflammatory bowel diseases have been excluded as marked in the list of exclusion criteria (line 400).

9. Please add the city for each device or kit (ThermoScientific, City, USA). As in line 337 The information about devices and kits have been adjusted accordingly (line 539).

10. Generally in methods part, authors mentioned some detailed analysis methods as in Metaproteomics and Metabolomics protocols while undetailed information in other as in coupled metagenomic and metatranscriptomic analyses etc please.

Thank you for this comment. The method part has been further extended and some processes have been explained in more detail, specifically regarding the biomolecular extractions (line 509).

11. I want to ask about the effect of drugs for treatment of both diseases (rheumatoid arthritis and Parkinson's) it does not mention by authors to study the effect of different drugs on microbiome composition in both diseases in cross-sectional and longitudinal study please clarify this important point.

Thank you for this important comment. In this clinical trial, we do not study the effects of the respective drugs; however, we are aware of findings from previous studies regarding the impact of the respective drugs on the gut microbiome and will take these into account during data analysis. The medication taken by the patients is recorded in REDCap.

12. Outcome : This section is adequate.

13. Discussion : Authors mainly talk about the microbiome dysbiosis and the effect of diet and fasting on microbiome and ignore discussing the interaction between microbiome and immune system in rheumatoid arthritis and Parkinson's please discuss this point deeply with introduction of more relevant new references.

We appreciate this valuable input of the reviewer. However, we have been advised by the BMJ Open Editorial office that standard journal formatting for protocols does not include a discussion as the manuscript comes without results. Thus, the Editorial office suggested us to delete the discussion except a few relevant parts that are now figuring in the introduction. Therefore, we cannot implement your comment. We hope you understand the circumstances.

14. Comments to the editor

I endorse the acceptance of the manuscript after performing the modifications.

We thank the reviewer for this positive final comment on our manuscript.

III. Reviewer 2: Prof. Marco Cosentino, University of Insubria

Comments to the Author:

The protocol describes a potentially interesting study on the effects of fasting on the gut microbiome and on immune parameters in RA and PD patients and in control subjects. After a cross-sectional analysis of the three groups, subgroups of RA and PD patients will be followed for 12 months during which they will practice prolonged fasting and time-restricted eating. The study is timely and relevant for the better understanding of the role of the relationship between immunity and the microbiome in the progression of autoimmune and inflammatory chronic diseases such as PD and RA.

Some issues however should be addressed to improve the description. Of the study.

Thank you very much for this positive comment on our study protocol and for the additional comments on the manuscript.

1. Throughout the text, the different groups of subjects should be homogeneously labelled to allow clear understanding of the protocol procedure. At present they are numbered but with different numbers in the various sections.

We appreciate this comment. The labelling has been unclear. The numbers in brackets in line 255 have been adjusted; the numbers in square brackets merely represent the number of patients included in each cohort: i.e., 60 patients with PD, 60 patients with RA and 60 healthy controls for cross-sectional part; 30 patients with PD and 30 patients with RA for the longitudinal study. We do apologize for the confusion.

2. Inclusion criteria include the diagnosis of PD or RA, but the diagnostic criteria should be better clarified to ensure a correct recruitment.

The appropriate diseases have been diagnosed by the patient's physicians according to the clinical practice guidelines of the national professional societies (Germany).

3. Exclusion criteria encompass antibiotic therapy as well as pregnancy and breast feeding. But what about such condition occurring during the study? Will those be reasons for exclusion? Why? Or why not? And what about high BMIs? While unlikely, they will be included? Why? And up to which values? Thank you for this comment. The exclusion criteria do only apply for including participants in the study on T0. The cross-sectional study is finished with T0 sample acquisition.

The question raised by the reviewer does only refer to the participants that will participate in the longitudinal study. No participant will be excluded during the study based on conditions such as antibiotic therapy, pregnancy, or breastfeeding. However, anamnesis and clinical information including medication will be carefully recorded in REDCap and all confounding factors will be considered during data analysis.

4. Pairing of patients with controls is said to be based on sex, age and education. What about BMI? What about comorbidities?

BMI criteria have been the same in exclusion / inclusion criteria for both groups, excluded comorbidities have been listed under the exclusion criteria (line 400).

5. Sample size is calculated for the cross-sectional but not for the longitudinal part of the study. While this might be acceptable, it should be justified.

We appreciate this important comment regarding sample size. The ExpoBiome cohort will consist of 180 adult individuals (line 609), these will be divided into three different groups: 60 patients with Parkinson's disease, 60 patients with rheumatoid arthritis and 60 healthy controls. For the longitudinal part, a subset of 60 adult individuals (30 patients with Parkinson's disease and 30 patients with rheumatoid arthritis) will be selected, based on their ability and willingness to participate in the longitudinal part of the study (12 months follow-up). The selected number of participants for the longitudinal study was based on feasibility due to the complexity and high costs of the clinical trial. The total number for subjects in the longitudinal study is smaller as each individual serves as his own control.

6. What about any procedures to check subject adherence to the study protocol during the longitudinal phase?

Thank you for pointing out this difficulty of clinical trials. The patients have numerous (line 359) visits during the longitudinal part. They will be requested to fill in questionnaires about their dietary behaviour and their well-being as well as the disease activity. In addition, blood samples are taken on each visit. Although the adherence of the patients cannot be profoundly controlled in the ambulatory setting, the blood samples might allow us to have additional insight into the nutritional habits as well as the fasting state of the patients on the day of the visit (blood glucose levels). Fortunately, our participants are very motivated and eager to comply with the study requirements.

7. The protocol states that no additional follow up will be performed on any subjects after the end of the study. However, no mention is made of any feedback information for participants about the study results. It would be polite and possibly useful to provide each person at least with a general idea of the outcomes of the study and whether it was successful, and which will be the implications of the obtained results, etc.

Thank you for this kind suggestion. While the participants of the longitudinal study will ultimately experience the effects of the intervention, the cross-sectional cohort will not receive immediate feedback, because the detailed analyses are time consuming. The participants will, however, be informed via social media channels (@wilmeslab) about future publications of the study results and any public available data.

VERSION 2 – REVIEW

REVIEWER	El Menofy, Nagwan Al-Azhar University, Microbiology and immunology
REVIEW RETURNED	17-Jun-2023

GENERAL COMMENTS	All modifications were performed as requested.
--

REVIEWER	Cosentino, Marco University of Insubria, Center for Research in Medical Pharmacology
REVIEW RETURNED	16-Jun-2023

GENERAL COMMENTS	The manuscript has been improved however several points raised in the first review round have been addressed only partially. First, I suggested that the diagnosis criteria used for PD or RA should be better clarified to ensure a correct recruitment. The authors answered that “the appropriate diseases have been diagnosed by the patient’s physicians according to the clinical practice guidelines of the national professional societies”. This is however in my opinion not appropriate. Please indicate the specific guidelines that were used for diagnosis as well as any eventual additional disease feature (e.g. clinical stage, for example for PD the H&Y stage). I also asked about the relevance of BMI and comorbidities in the pairing of patients and controls. The authors’ answer that “BMI criteria have been the same in exclusion / inclusion criteria for both groups, excluded comorbidities have been listed under the exclusion criteria (line 400)” actually does not address the confounding potential of not including BMI and comorbidities in the pairing procedure. What about the possibility that at the end of the day patients and controls will differ at baseline for these two critical features? Please, at least include in the text an explicit discussion about why or why not BMI and comorbidities will be taken into account, and the pros and cons, etc. The answer about sample size is acceptable. However please clarify also in the manuscript that no sample size calculation was performed for the longitudinal part of the study and the underlying reasons. Same for my question about any procedures to check subject adherence to the study protocol during the longitudinal phase: the
--

	answer is acceptable however the information should be also included in the appropriate section of the manuscript (the authors might consider that adherence to the study protocol is a critical issue also according to the reporting checklist). Finally, I suggested that participants should receive feed-back information about the study results, however the authors state that “the participants of the longitudinal study will ultimately experience the effects of the intervention” and that “the participants will, however, be informed via social media channels”. The first sentence apparently suggests that no information will be given to participants and that each subjects will be expected to understand just by him/herself about any effect, which of course is highly unlikely. The second sentence suggests that no targeted and accessible information will be provided to any participants. I kindly recommend the authors to consider that even the Declaration of Helsinki states that “All medical research subjects should be given the option of being informed about the general outcome and results of the study”.
--	--

VERSION 2 – AUTHOR RESPONSE

II. Reviewer 1 : Dr. Nagwan El Menofy, Al-Azhar University

Comments to the Author:

All modifications were performed as requested.

Thank you very much for accepting the modifications in the revised version of the manuscript.

III. Reviewer 2: Prof. Marco Cosentino, University of Insubria

Comments to the Author:

The manuscript has been improved however several points raised in the first review round have been addressed only partially.

Thank you very much for this positive comment and for the additional detailed comments on the manuscript. We have now addressed all the raised points and have made all requested modifications in the manuscript; detailed answers can be found below.

1. First, I suggested that the diagnosis criteria used for PD or RA should be better clarified to ensure a correct recruitment. The authors answered that “the appropriate diseases have been diagnosed by the patient’s physicians according to the clinical practice guidelines of the national professional societies”. This is however in my opinion not appropriate. Please indicate the specific guidelines that

were used for diagnosis as well as any eventual additional disease feature (e.g. clinical stage, for example for PD the H&Y stage).

Thank you very much for emphasizing the importance of the clarification of the diagnosis criteria for PD and RA. We have adapted the recruitment process as follows in line 279 in the revised manuscript:

“For PD, the patients were screened by an experienced movement disorders specialist for featuring at least two of resting tremor, bradykinesia, and rigidity according to the United Kingdom Parkinson’s Disease Society Brain Bank criteria¹. Additionally, patients must show evidence of a dopaminergic deficit, either with DaTScan imaging or with a clear response to dopaminergic drugs. Motor and non-motor symptoms are assessed with the MDS-UPDRS (part I – IV) including the Hoehn and Yahr (severity) scale. Additional PD-specific scales as PDSS2, PDQ39, NMSQ and NMSS are also used.

For RA patients, the diagnosis is made by an experienced RA specialist according to the European League Against Rheumatism (EULAR) criteria². The corresponding documentation is checked by the study physicians before including the patients into the ExpoBiome trial. All clinical stages of RA will be included. Additional RA-specific scales such as FFbH-R and joint status are used as well.”

¹Hughes AJ, Daniel SE, Blankson S, Lees AJ (1993) A clinicopathologic study of 100 cases of Parkinson’s disease. *Arch Neurol* 50, 140-148.

²Kay J, Upchurch KS. ACR/EULAR 2010 rheumatoid arthritis classification criteria. *Rheumatology (Oxford)*. 2012 Dec;51 Suppl 6:vi5-9. doi: 10.1093/rheumatology/kes279. PMID: 23221588.

(PDSS2 = Parkinson’s Disease Sleep Scale-2, PDQ39 = Parkinson’s Disease Questionnaire-39, NMSQ = Non-Motor Symptoms Questionnaire, NMSS = Non-Motor Symptoms Scale, FFbH-R = Funktionsfragebogen Hannover, joint status)

2. I also asked about the relevance of BMI and comorbidities in the pairing of patients and controls. The authors’ answer that “BMI criteria have been the same in exclusion / inclusion criteria for both groups, excluded comorbidities have been listed under the exclusion criteria (line 400)” actually does not address the confounding potential of not including BMI and comorbidities in the pairing procedure. What about the possibility that at the end of the day patients and controls will differ at baseline for these two critical features? Please, at least include in the text an explicit discussion about why or why not BMI and comorbidities will be taken into account, and the pros and cons, etc.

Thank you very much for highlighting possible confounding factors regarding the inclusion and exclusion criteria of our participants. Concerning the body mass index of the participants, for all cohorts we included a lower limit of BMI <18.5 as exclusion criteria, because this indicates

underweight, and fasting is in this case not recommended. We did, however, not include an upper limit in BMI as exclusion criteria. Fasting may be especially beneficial for patients with a BMI >24.9 and more than 60% of patients with RA are typically classified as overweight or obese¹. For comorbidities we excluded mainly diseases which are known to interfere with the gut microbiome and might be potential confounders: existing/current eating disorder (bulimia nervosa, anorexia nervosa) within the past 5 years, severe medical conditions: kidney deficiency with creatinine > 2mg/dl, and diagnosis of chronic inflammatory bowel diseases, celiac disease, or colorectal cancer according to the guidelines of the German Society of Gastroenterology. These exclusion criteria will optimize the pairing process of healthy controls and patients with either RA or PD. However, as we have two diseases with different anthropometric characteristics (including age, gender, BMI) and only one control group, adding additional inclusion and exclusion criteria in the recruitment process would compromise on optimized matching. Furthermore, for the longitudinal part of the study, each patient will serve as his/her own control over time. For additional confounding factors, especially in the cross-sectional study, multivariate statistical analysis will be performed. These factors will be accounted for by including confounders in the analysis, e.g., as covariate in the statistical models.

We have now added additional explanations in the main text as appropriate (lines 279, 503).

¹Feng, X., et al., Body Mass Index and the Risk of Rheumatoid Arthritis: An Updated Dose-Response Meta-Analysis. *Biomed Res Int*, 2019. 2019: p. 3579081.

3. The answer about sample size is acceptable. However please clarify also in the manuscript that no sample size calculation was performed for the longitudinal part of the study and the underlying reasons.

We appreciate the reviewer's valuable comment. As suggested, we have now included an explanation and a statement about the sample size in the main text in line 478 as follows:

"For the longitudinal part, a subset of 60 adult individuals (30 patients with Parkinson's disease and 30 patients with rheumatoid arthritis) are selected, based on their ability and willingness to participate in the longitudinal part of the study (12 months follow-up). The selected number of participants for the longitudinal study is based on feasibility due to the complexity and high costs of the clinical trial. The total number for subjects in the longitudinal study can be smaller, as each individual serves as their own control (longitudinal study: 10 additional visits over a period of 12 months).

4. Same for my question about any procedures to check subject adherence to the study protocol during the longitudinal phase: the answer is acceptable however the information should be also included in the appropriate section of the manuscript (the authors might consider that adherence to the study protocol is a critical issue also according to the reporting checklist).

Thank you very much for pointing out this critical difficulty of clinical trials. As suggested by the reviewer, we also included an explanation and a statement about the adherence to the study protocol in the main text in line 324 as follows:

“Although the adherence of the patients cannot be profoundly controlled in the ambulatory setting, the blood samples will allow us to have additional insight into the nutritional habits as well as the fasting state of the patients on the day of the visit (blood glucose levels).”

5. Finally, I suggested that participants should receive feed-back information about the study results, however the authors state that “the participants of the longitudinal study will ultimately experience the effects of the intervention” and that “the participants will, however, be informed via social media channels”. The first sentence apparently suggests that no information will be given to participants and that each subjects will be expected to understand just by him/herself about any effect, which of course is highly unlikely. The second sentence suggests that no targeted and accessible information will be provided to any participants. I kindly recommend the authors to consider that even the Declaration of Helsinki states that “All medical research subjects should be given the option of being informed about the general outcome and results of the study”.

Thank you for pointing out this important fact that we gladly adopt to the protocol. According to the suggestions, the participants of the study will be informed after the completion of the analysis in the dissemination phase by the respective recruitment site. The following section has been added in the Ethic and dissemination section in the main text in line 549 (the abstract has a 300 word limitation; therefore we kept a short version in the abstract).

“Study participants will be contacted and informed by the respective clinical sites about the outcome and results of the study, once the data analysis has been completed (dissemination phase).”

In addition, study results will be made available to a broad public via press-releases and through social media channels.

VERSION 3 – REVIEW

REVIEWER	Cosentino, Marco University of Insubria, Center for Research in Medical Pharmacology
REVIEW RETURNED	13-Jul-2023
GENERAL COMMENTS	The authors finally addressed the various remarks in a satisfying manner.

VERSION 3 – AUTHOR RESPONSE

Reviewer: 2

Prof. Marco Cosentino, University of Insubria

Comments to the Author:

The authors finally addressed the various remarks in a satisfying manner.

Reviewer: 2

Competing interests of Reviewer: None.

We thank the reviewers for their constructive criticism and help to get the revised manuscript in an acceptable format.